# Electron density-based GPT for optimization and suggestion of host–guest binders

Juan M. Parrilla-Gutiérrez[1,2,4], Jarosław M. Granda [1,3,4], Jean-François Ayme[1,4], Michał D. Bajczyk[1], Liam Wilbraham[1] & Leroy Cronin [1]✉

Here we present a machine learning model trained on electron density for the production of host–guest binders. These are read out as simplified molecular-input line-entry system (SMILES) format with >98% accuracy, enabling a complete characterization of the molecules in two dimensions. Our model generates three-dimensional representations of the electron density and electrostatic potentials of host–guest systems using a variational autoencoder, and then utilizes these representations to optimize the generation of guests via gradient descent. Finally the guests are converted to SMILES using a transformer. The successful practical application of our model to established molecular host systems, cucurbit[$n$]uril and metal–organic cages, resulted in the discovery of 9 previously validated guests for CB[6] and 7 unreported guests (with association constant $K_a$ ranging from 13.5 M$^{-1}$ to 5,470 M$^{-1}$) and the discovery of 4 unreported guests for [Pd$_2$1$_4$]$^{4+}$ (with $K_a$ ranging from 44 M$^{-1}$ to 529 M$^{-1}$).

The chemical space of synthetically accessible molecules is vast[1]. Navigating this space efficiently requires computational-based screening techniques such as deep learning[2] to fast track the discovery of compounds of interest[3,4]. The use of algorithms for chemical discovery, however, requires the translation of molecular structures into digital representations that are usable by a computer[5], and the development of algorithms operating on these representations to generate new molecular structures[6]. Strings of characters, such as the simplified molecular-input line-entry system (SMILES), where molecules are represented in 'words'—for example, 'C1C=C1' (cyclopropene)—are among the most widespread digital representations of molecules. Using state-of-the-art natural language processing, these representations are directly compatible with artificial intelligence techniques, such as recurrent neural networks[7] or the transformer model[8,9]. As artificial intelligence performs better using continuous data, SMILES strings have also been converted into continuous latent representations[10]. Furthermore, molecules have been digitized into graphs compatible with modern graph neural networks[11–13], or as three-dimensional (3D) shapes—by extending a volume around the sparse atoms using a wave function[14], or by using density functional theory to generate an electron density[15,16]

treated as a 3D volume[17]. In this regard, it is important to note that the Hohenberg–Kohn theorems state that the energy of an atomic system is unambiguously determined by the electron density of the system. In addition, the electron density delivers the lowest energy if and only if the input density is the true ground-state density[18].

The representation of molecules as 3D volumes has the advantage of enabling the application of the latest artificial intelligence techniques, such as convolutional neural networks[19]. So far, most applications of 3D volumes as molecular descriptors are focused on predicting properties[20], or de novo drug design[21]. However, the utilization of a 3D volume as molecular descriptors is currently hindered by the absence of an efficient method to correlate these volumes with clear molecular structures. Over the past 40 years, host–guest systems have been increasingly studied due to the propensity of molecular containers—hollow organic molecules or hollow supramolecular architectures—to alter the chemical and physical properties of molecules by sequestering them from the bulk phase in their cavities[22]. Host–guest systems have found a wide range of applications, from catalysis[23,24] to biomedical engineering[25,26], materials science[27] and the stabilization of reactive molecules[28]. Cucurbit[$n$]urils and metal–organic cages are among the

[1]School of Chemistry, University of Glasgow, Glasgow, UK. [2]School of Computing, Engineering and Built Environment, Glasgow Caledonian University, Glasgow, UK. [3]Institute of Organic Chemistry, Polish Academy of Sciences, Warsaw, Poland. [4]These authors contributed equally: Juan M. Parrilla-Gutiérrez, Jarosław M. Granda, Jean-François Ayme. ✉e-mail: lee.cronin@glasgow.ac.uk

most successful designs of molecular containers. Cucurbit[*n*]urils are donut-shaped molecules composed of *n* glycoluril units connected via methylene bridges. They are characterized by a hydrophobic central cavity gated by two sets of dipolar carbonyl moieties, enabling them to bind neutral and cationic species[29,30]. Metal–organic cages are discrete hollowed 3D structures generated by the self-assembly of polytopic ligands around metal cations[22,31–33]. Lantern-shaped cages are a notable example of such containers. They are assembled via the coordination of four ditopic 'banana-shaped' ligands around two Pd(II) ions[34], creating an (often hydrophobic) cavity capable of binding charged or neutral aromatic guests in various organic solvents[35,36]. Although host–guest chemistry has had notable achievements, the discovery of unreported guests for existing systems or the optimization of new host–guest systems remains a laborious and costly iterative process, impeding the pace of scientific advancement.

Here we demonstrate that representing host molecules as 3D volumes (that is, as electron density decorated with electrostatic potential) enables the computer-aided discovery of guests for this host without having any knowledge of the host–guest system besides the chemical structure of the host (Fig. 1). In doing so, we establish that a transformer model can be trained to efficiently convert 3D volume molecular descriptors into SMILES representations, generating defined molecular structures that are usable in real-world applications by an expert chemist. We also establish that molecules can be efficiently represented as 3D volumes by decorating their electron densities with electrostatic potential data[37] and that these two features are sufficient to inform the discovery of guest molecules for a host by optimizing the volumetric shape and charge interactions between their 3D descriptors using an autoregressive sampling scheme[38]. We experimentally verified our workflow by generating both literature-validated and unreported guests for two well-known and studied host–guest systems: a cucurbit[*n*]uril and a metal–organic cage.

## Results

### Rational and workflow overview

The computer-aided discovery of experimentally validated guests for the cucurbituril **CB[6]** and for the metal–organic cage $[Pd_2\mathbf{1}_4]^{4+}$ (**1** refers to 1,3-bis(pyridin-3-ylethynyl)benzene) required a two-tier workflow (Fig. 1). First, an in silico workflow was devised to generate virtual libraries of potential guest molecules for these two hosts (Fig. 1a). Then an in vitro workflow was put in place, which involved the selection of the most promising guest candidates from these virtual libraries by an expert chemist for experimental testing (Fig. 1b). The in silico generation of guest molecules for **CB[6]** and $[Pd_2\mathbf{1}_4]^{4+}$ was achieved through the workflow depicted in Fig. 1a, which consisted of the following steps. (1) A training set of 3D electron density volumes was derived from the molecules in the publicly available QM9 dataset—a chemical space containing over 130,000 small molecules with up to 9 heavy atoms (C, O, N and F). Then a 'molecule generator' was created by modeling this training set of 3D electron density volumes using a variational autoencoder (VAE; Fig. 1a), thus allowing for the generation of 3D electron density volumes beyond those derived from the QM9 dataset[39]. This VAE molecule generator operates by encoding 3D electron density volumes into a one-dimensional (1D) latent space and then generating 3D electron density volumes corresponding to molecules by decoding from this 1D latent space. Interestingly, this approach only generated chemically plausible molecules. (2) Our VAE molecule generator and a gradient-descent optimization algorithm were used to generate a library of guest molecules—in the form of 3D electron density volumes—for a given host molecule. Guest molecules were generated by minimizing the overlap between the host and guest electron densities while optimizing their electrostatic interactions. (3) As it can be challenging for human operators to convert 3D electron density volumes into chemically interpretable structures, a transformer model was trained to translate these volumes into SMILES representations,

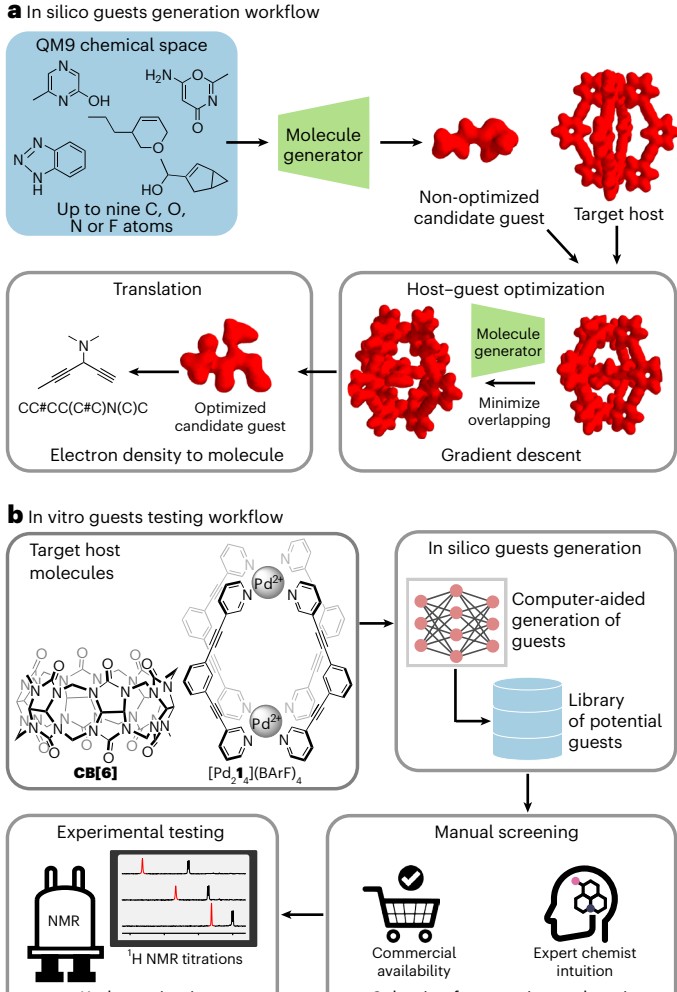

**a** In silico guests generation workflow

QM9 chemical space

Up to nine C, O, N or F atoms

Molecule generator

Non-optimized candidate guest

Target host

Translation

CC#CC(C#C)N(C)C Optimized candidate guest

Electron density to molecule

Host–guest optimization

Molecule generator

Minimize overlapping

Gradient descent

**b** In vitro guests testing workflow

Target host molecules

**CB[6]** $[Pd_2\mathbf{1}_4](BArF)_4$

In silico guests generation

Computer-aided generation of guests

Library of potential guests

Experimental testing

NMR

$^1$H NMR titrations

$K_a$ determination

Manual screening

Commercial availability

Expert chemist intuition

Selection for experimental testing

**Fig. 1 | Discovering novel guest molecules through electron density volumetric representation. a**, The QM9 chemical space (with C, O, N and F referring to carbon, oxygen, nitrogen and fluorine, respectively) was used to train our VAE. Once trained, the latent space created by the VAE (a 1D space) could be navigated, and the 3D structural information of a target molecule was reconstructed using the VAE decoder (molecule generator). Navigating the latent space created, the 3D structural information of a target molecule (molecule generator) was reconstructed using the VAE. Given a target host, gradient descent was used to discover guests that maximize the electrostatic interactions with the host, while minimizing electron density overlap. The 3D volumes of the candidate guests were translated into SMILES, giving the full chemical information required for their synthesis. **b**, The potential guest molecules generated by the optimization algorithm for cucurbituril **CB[6]** and metal–organic cage $[Pd_2\mathbf{1}_4]^{4+}$ were selected by an expert chemist for experimental testing based on their structural resemblance with known guests and, second, their commercial availability. The $K_a$ of the guest molecules selected for **CB[6]** or $[Pd_2\mathbf{1}_4]^{4+}$ was quantified by direct $^1$H NMR titration.

capturing all necessary information required to describe molecules in a format that is more easily understood by expert chemists. Following the in silico generation of potential guest molecules for **CB[6]** and $[Pd_2\mathbf{1}_4]^{4+}$, an in vitro workflow was put in place to experimentally test the most promising candidates.

The following describes the experimental process used (Fig. 1b). (1) The guests generated by our in silico workflow for **CB[6]** and for $[Pd_2\mathbf{1}_4]^{4+}$ (Fig. 1b) were triaged by an expert chemist for experimental testing. Promising guests for testing were selected based on their structural resemblance with known guests for **CB[6]** or $[Pd_2\mathbf{1}_4]^{4+}$, the intuition of the expert chemist and their commercial availability.

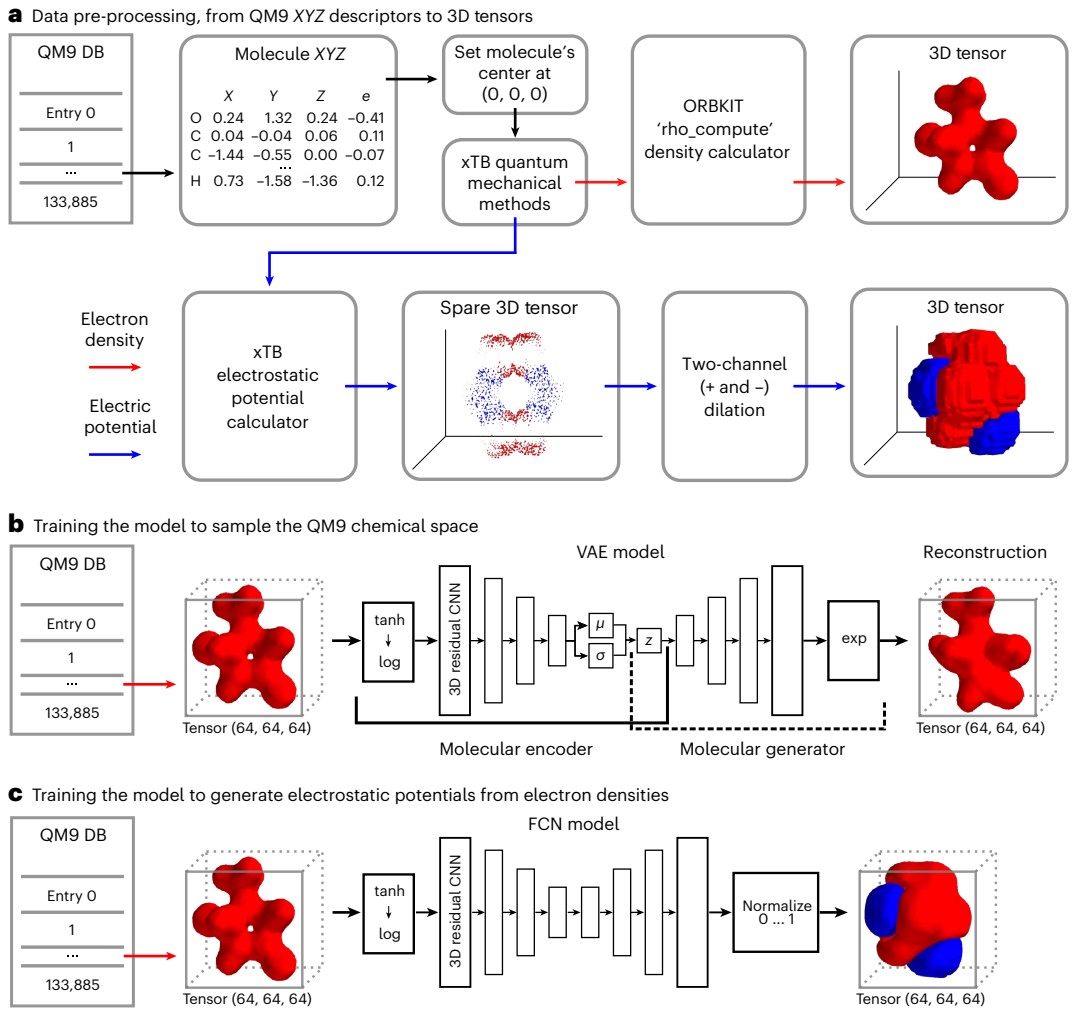

**Fig. 2 | Sampling the QM9 chemical space using a VAE. a,** Conversion of the QM9 dataset (DB) in *XYZ* format (*XYZ* values are shown solely for representation purposes) to electron densities and electrostatic potentials using quantum mechanical methods and density calculators. xTB refers to the Semiempirical Extended Tight-Binding Program Package software; *e* refers to partial charges on each atom. **b,** Training a VAE to model the QM9 chemical space. The encoder side of the VAE was used to encode molecules into their 1D latent representations, while the decoder side of the VAE was used to generate molecules given 1D latent vectors. Molecules were generated into a 3D tensor of 64 units (voxels) per side. $\mu$, $\sigma$ and $z$ refer to mean, standard deviation and latent space, respectively. **c,** Utilizing an FCN network to calculate the electrostatic potential of a molecule given its electron density. tanh → log refers to the fact that each element in the input tensor was put through a tanh operation followed by a log operation. CNN, convolutional neural network.

(2) The affinity of the guests selected for **CB[6]** or $[Pd_2\mathbf{1}_4]^{4+}$ was quantified by direct $^1H$ NMR titration. Notably, the guests generated in silico contained a mixture of molecules previously known to bind to the host (or closely related) and molecules defying the intuition of the expert.

**Modeling and sampling the QM9 chemical space**

The QM9 dataset was chosen as a subset of the chemical space for this study. Among different properties, the QM9 dataset provides for each molecule its *XYZ* coordinates and its SMILES representation. The data pre-processing started by converting each QM9 molecule from its *XYZ* coordinates into a 3D grid representing its isosurfaces as electron densities at each location (Supplementary Sections 1.1, 1.2 and 1.3). The electron density grid of each molecule was used to calculate its 3D electrostatic potential using quantum methods (Fig. 2a). Once the electron density grid was generated for each molecule, it was used to train a VAE (Fig. 2b and Supplementary Sections 1.4 and 1.5). Using a VAE for this task guarantees four key features: (1) a molecule encoder, generating a unique 1D latent representation of any molecule's electron density fitting inside the 3D tensor defined earlier, (2) a molecule similarity check so that similar molecules are encoded using similar

latent vectors, (3) a molecule generator, generating a 3D electron density tensor from any 1D latent representation, and (4) a chemical plausibility check, guaranteeing that any molecule generated from the latent vector is chemically plausible. A fully convolutional neural (FCN) network was then used to generate the electrostatic potential volume from the corresponding electron density volume (Fig. 2c and Supplementary Section 1.6).

**Translating electron densities into SMILES**

A transformer model was used to translate the 3D electron density tensors generated into SMILES describing the molecules fitting the closest to these volumes (Fig. 3 and Supplementary Sections 1.10, 1.11 and 1.12), thus enabling the identification of clear molecular targets exploitable by chemists from the abstract 3D tensor generated. The inner workings of our transformer model followed the standard implementation[9] (Fig. 3a). Our focus was placed on designing embedding layers to transform the 3D electron densities into 1D latent sequences. The transformer's encoder received as input 3D tensors such as the ones shown in Fig. 2a, and the transformer's decoder received tokenized SMILES sequences. While the decoder's input used a standard 'token

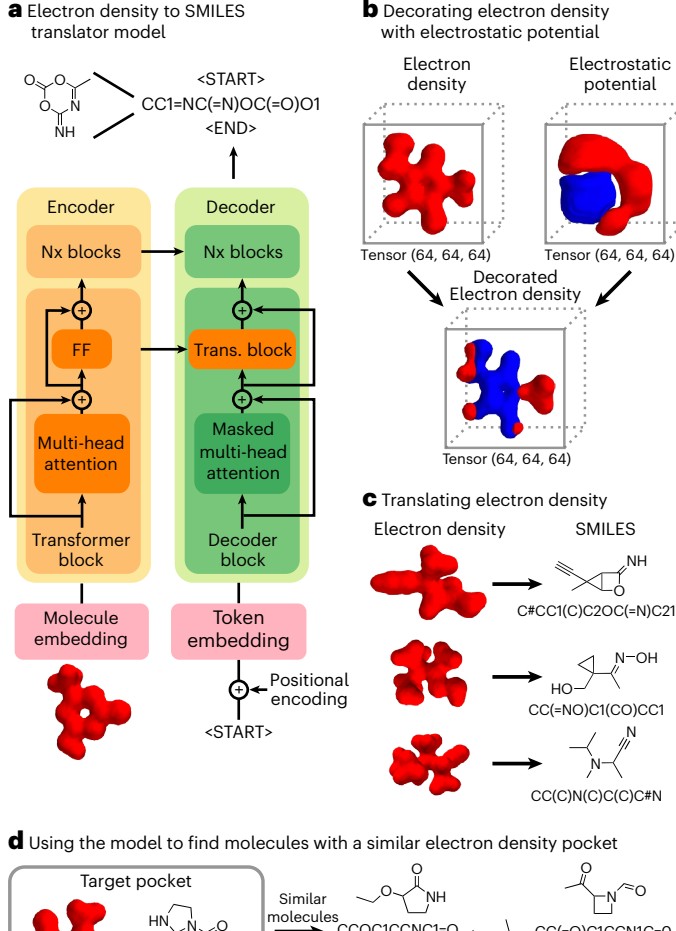

**a** Electron density to SMILES translator model

**b** Decorating electron density with electrostatic potential

**c** Translating electron density

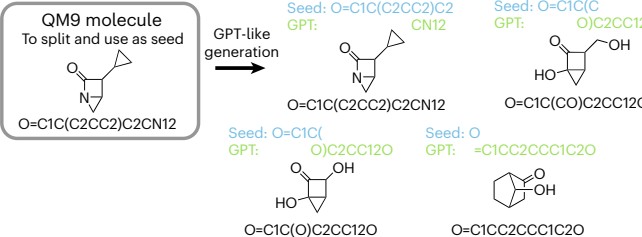

**d** Using the model to find molecules with a similar electron density pocket

**e** Using the model as a SMILES GPT generator

**Fig. 3 | Transforming electron densities into SMILES representations using a transformer model followed by optimization of the guests for a target host via gradient descent. a**, Inputs of either decorated or non-decorated electron densities. FF, fully connected feed-forward network; Trans., transformer; Nx refers to the blocks being repeated (or stuck) *N* times. **b**, Standard implementation of the transformer model to design a molecule embedding layer transforming 3D volumes into 2D tensors later usable in the different attention mechanisms. In the electrostatic potential tensor, areas in red represent areas with positive electrostatic potential while areas in blue represent areas with negative electrostatic potential. **c**, Examples of different translated electron densities. **d**, Implementation of using the probabilities outputted by the last softmax layer to randomly sample one of the tokens, allowing for finding molecules that fit a defined 3D cavity. **e**, Behavior of the transformer as a GPT model working with SMILES, when the encoder is disabled.

embedding layer', the embedding layer from the encoder had to transform 3D molecules into two-dimensional (2D) attention matrices so that it could be operated with the decoder's attention matrices. To do so, the input 3D data first had to be transformed and expanded

into four dimensions (Tensorflow's 3D convolution layer requires the input data to be four-dimensional (4D)) before these 4D data were transformed into 2D.

The transformation from 3D to 4D was achieved using two different strategies. Initially, electron density 3D tensors were simply expanded into four dimensions (Fig. 2a). Later, to facilitate the translation from 3D tensor to SMILES, electron density 3D tensors were decorated with their related electrostatic potentials (Fig. 3b) before being expanded into four dimensions. The transformation of the 4D tensors into 2 dimensions, was achieved using convolutions with filters set to 1 to squeeze out these dimensions. Using the test set as reference, and using decorated electron density, our transformer model perfectly predicted its SMILES representations with a 98.125% accuracy (Fig. 3c). Individual tokens were predicted with a 99.114% accuracy. Setting the decoder to choose the next token using probability-based sampling could be used to find molecules with a similar pocket to a target molecule (Fig. 3d). The transformer's decoder could also be isolated to be a purely generative model like GPT (Fig. 3e).

**Discovering and optimizing guests for a given host molecule**
Our VAE, FCN and transformer model were implemented to enable the generation of guest molecules solely knowing the electron data of a target host (Figs. 4 and 5). This task was tackled as an optimization problem (Supplementary Section 2). Given a host, gradient descent was used to find guests using a combination of three fitness functions (Fig. 4a): (1) the molecular size of the molecule should be maximized; (2) the overlapping between the electron densities of a host and a guest should be minimized (for a guest to fit inside the host's cavity their electron densities cannot overlap); and (3) the electrostatic interactions between a host and a guest should be maximized (their electrostatic potentials should be inversely aligned to increase their possible binding—the positive regions of the host should be near negative regions of the guest, and vice versa).

Before starting the optimization pipeline, a random population of guests had to be created (Fig. 4b). To do so, we initially generated random latent vectors, used the VAE molecule decoder to generate the corresponding 3D molecules, and then used the FCN to calculate their electrostatic potentials. Our optimization pipeline operates as follows: (1) given a latent representation, the VAE is used to obtain its corresponding 3D volume tensor, (2) from this tensor, the FCN is used to calculate the electrostatic potential (if required), (3) then in the 3D space, the fitness value the molecule is calculated against the target fitness function (for example, how much they overlap), and (4) the fitness value obtained informs the modification of the latent vector using a gradient descent.

The size of the molecules was optimized first, guaranteeing that some overlap exists between the host and the guest (Fig. 5a). For **CB[6]**, this step was not needed, because the initial random guests already overlap with it; however, for $[Pd_2\mathbf{1}_4]^{4+}$, this step was required as the initial random guests were smaller than the cavity of the cage. Next, the overlapping between host and guest was optimized (minimized) while optimizing (maximizing) their electrostatic interactions (Fig. 5b). As these two optimization functions aimed to do opposite things—one tried to decrease the size of the molecule, while the other tried to increase it—they were combined into a single function where the ratio between them could be chosen. These two steps were iterated until the fitness values plateaued, after which the resulting optimized guests were translated into SMILES using our transformer model (Fig. 5c).

**Quantitative study of the host–guest recognition**
**Study of the cucurbituril CB[6] system.** With its cavity of 3.9 Å in diameter at its narrowest, **CB[6]** (Fig. 6a) is the most common of the cucurbiturils[30]. In aqueous formic acid ($HCO_2H/H_2O$ 1:1, v/v), it has been shown to only weakly associate with aliphatic alcohols, acids and nitriles[40] but to form strong 1:1 inclusion complexes with derivatives of

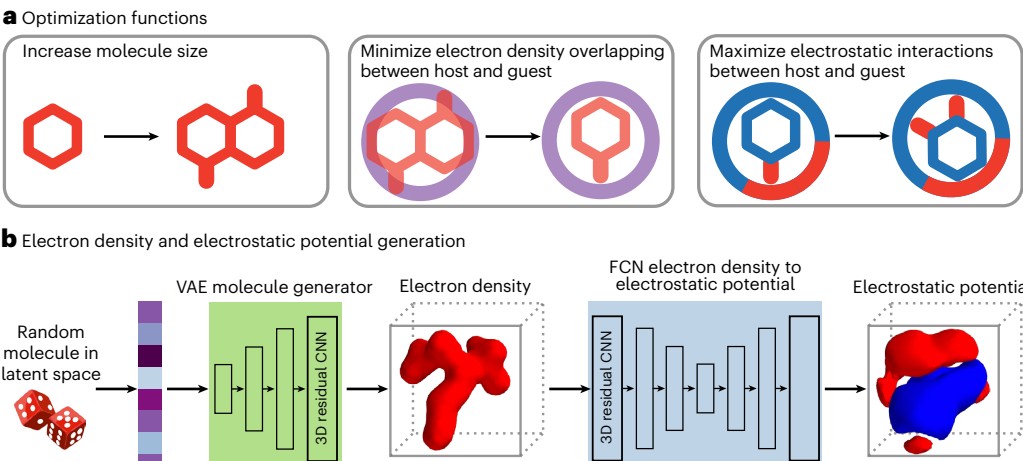

**a** Optimization functions

Increase molecule size

Minimize electron density overlapping between host and guest

Maximize electrostatic interactions between host and guest

**b** Electron density and electrostatic potential generation

Random molecule in latent space

VAE molecule generator

3D residual CNN

Electron density

Tensor (64, 64, 64)

FCN electron density to electrostatic potential

3D residual CNN

Electrostatic potential

Tensor (64, 64, 64)

**Fig. 4 | Optimizing guests for a target host via gradient descent. a**, Targeting of multiple fitness functions for optimizing host–guest interactions: maximize the size of the guest, minimize its overlapping with the host and maximize its electrostatic interactions. In the right panel, areas in red represent areas with positive electrostatic potential while areas in blue represent areas with negative electrostatic potential. **b**, Initial population of guests generated through random sampling. Using random sampling, a 1D vector in the latent space was generated. Via the VAE, a 3D electron density could be reconstructed from this 1D vector. From this 3D electron density, and using the FCN, its electrostatic potentials were calculated.

primary and secondary ammonium salts[29]. In the former, the formation of the host–guest complex is (mainly) driven by hydrophobic effects (notably, via the liberation of 'high-energy water' molecules) whereas in the latter both hydrophobic effects and ion–dipole interactions (between the ammonium cation and the carbonyl groups of the **CB[6]**) play a role[30]. The importance of both electronic and steric considerations in the binding of guests with **CB[6]** and the fact that most known guests associating with **CB[6]** are smaller than ten heavy atoms make this cucurbituril an appropriate choice for testing our optimization algorithm.

Our algorithm generated nine previously known guests for **CB[6]**, validating our approach. The affinity of **CB[6]** for **G¹–G⁹** (guests 1 to 9) was previously reported in the literature, with association constant ($K_a$) values ranging from 18 M$^{-1}$ to $10^5$ M$^{-1}$ in HCO$_2$H/H$_2$O 1:1 v/v (Fig. 6a). Our algorithm also identified seven potential new guests for **CB[6]**, which our expert chemist deemed worthy of experimental testing. The affinity of **CB[6]** for these new guests was evaluated via direct $^1$H NMR titration in HCO$_2$H/H$_2$O 1:1 v/v (Supplementary Section 3.3). In all seven cases, a single set of signals was observed for the host–guest system, indicating that the system is in fast exchange on the NMR timescale. Upon complexation, the resonance of the aliphatic chains of the guest molecules were shifted upfield, indicating their encapsulation within the **CB[6]** cavity. The association constants of **G¹⁰–G¹⁶** with **CB[6]** were found to follow previously established trends[29], spanning from 13.5 M$^{-1}$ to 5,470 M$^{-1}$ (Fig. 6a). Linear secondary amines **G¹⁰** and **G¹¹** gave two of the highest association constants measured, with **G¹⁰** having the highest association constant due to its longer alkane chain[29]. Branched alkylamine **G¹²–G¹⁶** bound moderately with **CB[6]**. The monomethylation of the amine of **G¹³** had little influence on its interaction with **CB[6]** as both **G¹²** and **G¹³** had similar $K_a$. Despite ethyl-substituted *n*-alkylamine reportedly being unable to form inclusion complexes with **CB[6]**[29], **G¹⁴** was found to be bound moderately by **CB[6]**.

**Study of cage [Pd₂1₄](BArF)₄ system.** Compared with **CB[6]**, [Pd₂**1**₄]$^{4+}$ (Fig. 6b) allowed us to test our optimization algorithm in more demanding circumstances: (1) the bigger cavity size of [Pd₂**1**₄]$^{4+}$ means that most known binders of the cage are bigger than ten heavy atoms and (2) binding neutral guests in organic solvents is inherently more challenging than binding charged guests in water (neutral guests have to compete with the anions associated with the cationic cage for its

cavity and solvophobic effects are less favorable in organic solvents than in water)[36]. For our study, the non-coordinating anion tetrakis[3,5-bis(trifluoromethyl)phenyl]borate (BArF) was selected as a counteranion for the cage to maximize the availability of the inner cavity of the cage to charge-neutral guests by minimizing ion pairing[36].

For [Pd₂**1**₄]$^{4+}$, the optimization algorithm generated only unknown guest molecules (Fig. 6b). Compared with **CB[6]**, featuring a cavity with a diameter of approximately 3.9 Å (ref. 30), [Pd₂**1**₄]$^{4+}$ has a notably larger cavity, measuring approximately 6 Å in width and 10 Å in depth[35]. This increased cavity size led our model to generate larger guest molecules, resulting in very few of them being commercially available, thereby limiting the pool of molecules available for experimental testing. The strength of binding between four potential unreported guests and the [Pd₂**1**₄](BArF)₄ was tested via direct $^1$H NMR titration in CD₂Cl₂ (Supplementary Section 3.4). In all cases, the host–guest system was in fast exchange on the NMR timescale. Upon addition of the guests to the cage, a unique set of signals was observed by $^1$H NMR spectroscopy. This set of signals differed substantially from a mere superimposition of the spectra of the individual species. Notably, the signals from the cage showed a downfield shift, providing compelling evidence of the successful encapsulation of the guest molecule within the cage. In all four cases (Fig. 6b), the affinity of the guest for [Pd₂**1**₄](BArF)₄ was in line with the lower range of affinities previously reported for 'small-sized neutral guests' in CD₂Cl₂ (that is, guest formed of ten heavy atoms or fewer, such as **G¹⁹**)[36]. The lack of 'strong binders' in the molecules tested could be attributed to the fact that the cavity size of [Pd₂**1**₄](BArF)₄ pushes the limits of our model and workflow capabilities: (1) as previously highlighted, the scarcity of commercially available options within the dataset generated by our model hampered the quality of the guest tested, and (2) all known strong binders for [Pd₂**1**₄]$^{4+}$ feature an aromatic core substituted by two donor groups *para* to each other[35,36]. Apart from **G¹⁷**, this structural feature inherently increases the size of the molecule beyond the ten-heavy-atoms limit of our model. Such size constraint on the molecules generated by our model stems from the utilization of QM9 for its training, making it unlikely to generate molecules that exceed ten heavy atoms in size. Importantly, **G²¹–G²⁴** demonstrate that the optimization algorithm was capable of generating guests with (1) the right hydrogen-bond acceptor groups (the cage having no affinity for fully hydrocarbon guests, such as *p*-xylene or naphthalene)[35] and (2) the right rigidity (the cage having no affinity for flexible guests, such

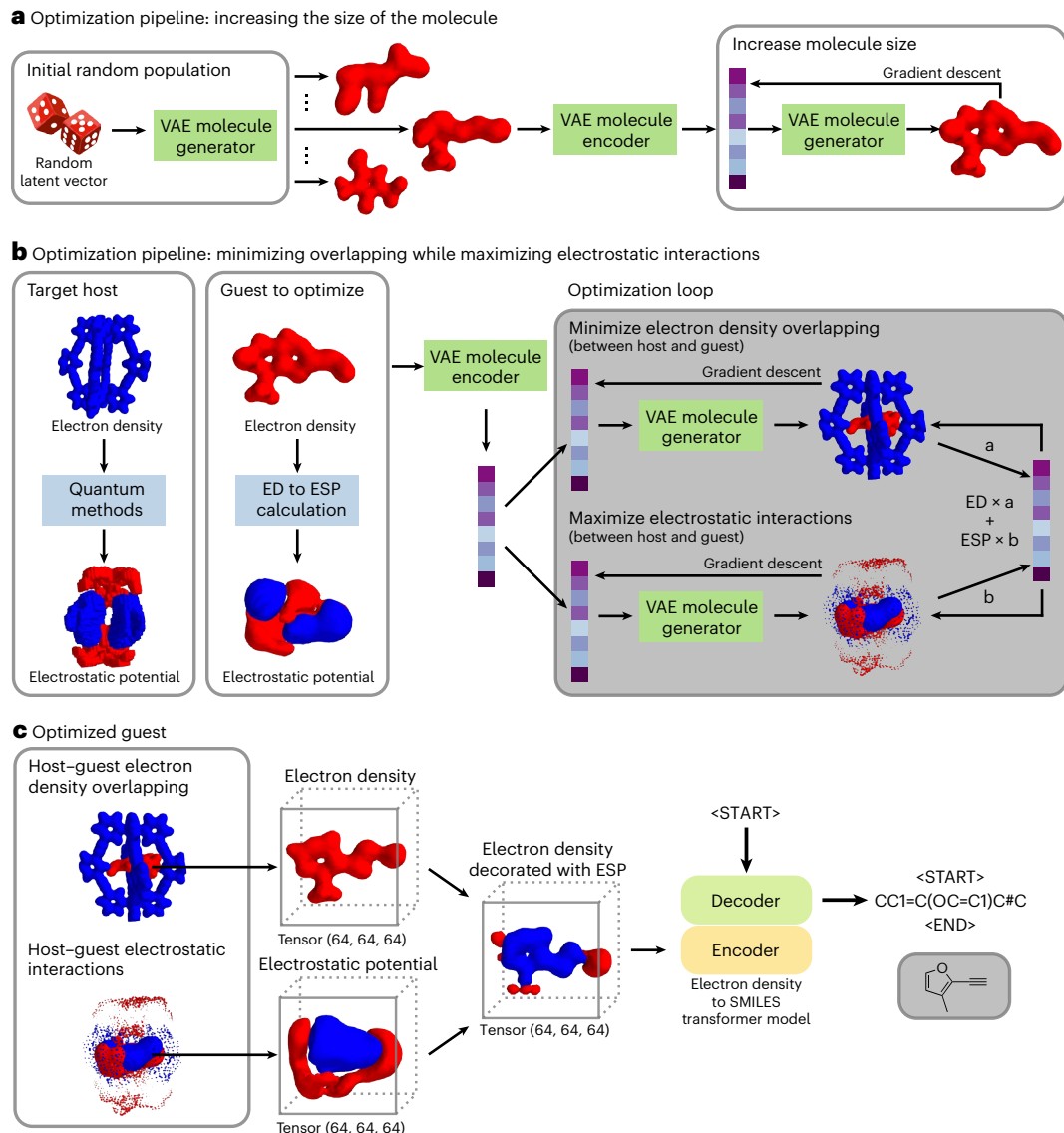

**a** Optimization pipeline: increasing the size of the molecule

**b** Optimization pipeline: minimizing overlapping while maximizing electrostatic interactions

**c** Optimized guest

**Fig. 5 | Optimization pipeline and generation of SMILES representations of the guests. a**, Optimization pipeline to maximize guest size. **b**, Optimization pipeline simultaneously minimizing host–guest electron density overlapping while maximizing its electrostatic interactions. ED, electron density; ESP, electrostatic potential. In the electrostatic potential tensor, areas in red represent areas with positive electrostatic potential while areas in blue represent areas with negative electrostatic potential. **c**, Use of our transformer model to obtain the SMILES representation of the guest generated.

as 1,4-dicyanobutane or 1,6-dicyanohexane)[35,36]. The lack of molecules containing two donor groups generated by the optimization algorithm could be (in part) attributed to the molecule size limitation imposed by the use of QM9 to train the algorithm (most known guests with two donor groups being ten heavy atoms or bigger, such as **$G^{18}$**).

## Discussion

While our research focused on using SMILES notation to represent molecules, we also tested other similar formats, such as Self-referencing Embedded Strings (SELFIES)[41] (Supplementary Sections 2.3.10 and 2.3.11). Even though SELFIES has the advantage of being a 100% robust molecular string representation, it did not improve our results. Although the QM9 dataset contained molecules of perfect size to be guests of a host such as **CB[6]**, a limitation we encountered during this research is that the metal–organic cage $[Pd_2\mathbf{1}_4]^{4+}$ had a bigger cavity, requiring bigger guest molecules. We overcame this limitation by adding a function that increased the size of the molecules as much as possible, but in future research we aim to use a dataset that contains bigger molecules, such as the GDB-17 dataset[42]. Later, we aim to embed the selection of new ligands into the generative process[43,44], with the objective of synthesizing the molecules autonomously on an automated synthetic platform, such as a Chemputer robot[45], closing the loop between optimization and testing, creating a cyber-physical closed loop system.

## Methods
### Source code libraries

The source code developed in this research was written using Python 3.9. The machine learning models were written using Tensorflow. Most of the development and testing was done using Tensorflow 2.7. In later stages, we updated our code Tensorflow to version 2.10. We have tested our code with the latest version available at the moment of writing this paper (2.13), but this version did not work with some of our scripts. We used Conda to create and handle the Python environment. Within our source code, two Conda environments are provided: one for Tensorflow 2.7 and one for Tensorflow 2.10. See Supplementary Sections 1.1 and 1.2.

**a** Host–guest study of cucurbituril **CB[6]**

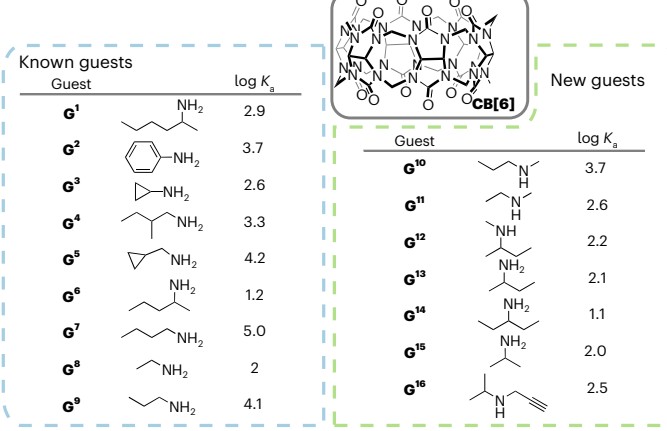

**b** Host–guest study of cage [Pd₂**1**₄](BArF)₄

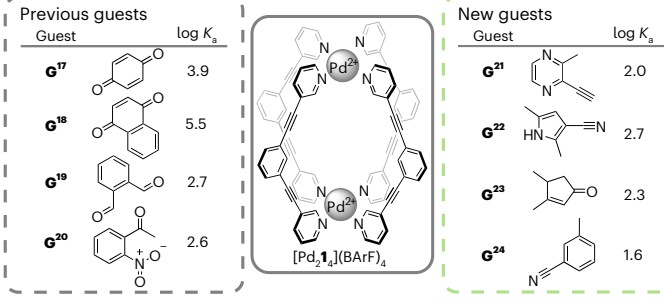

**Fig. 6 | Optimized and previously known guests for CB[6] and optimized guests for [Pd₂1₄]⁴⁺.** a, Structures and log $K_a$ values for guest molecules generated by the optimization algorithm for **CB[6]** and the structure of **CB[6]**. Association constants were measured in HCO₂H/H₂O 1:1 v/v. The association constants between **CB[6]** and guests 1 to 9 (**G¹**–**G⁹**) in HCO₂H/H₂O 1:1 v/v were previously reported in the literature[29]. **b**, Left: structures and log $K_a$ values for guest molecules previously reported in the literature for [Pd₂**1**₄](BArF)₄; association constants were measured in CD₂Cl₂ (ref. 36; these four guests were not generated by our model). Middle: the structure of [Pd₂**1**₄]⁴⁺. Right: structures and log $K_a$ values for guest molecules generated by the optimization algorithm for [Pd₂**1**₄](BArF)₄. Association constants were measured in CD₂Cl₂.

## Generating the training dataset

This research used the publicly available QM9 dataset from ref. 38. This dataset contains 133,885 molecules of up to 9 heavy atoms (carbon, oxygen, nitrogen and fluorine). For each molecule, this dataset contained different data entries. This research focused on their SMILES representations and the *XYZ* information. Within our source code, we have prepared a script that downloaded the dataset, generated the electron densities and electrostatic potentials for all the molecules present, and saved them into a Tensorflow's TFRecord file (of size 240 Gb). This command can be executed as '$ python bin/dataset/generate_dataset.py QM9'.

This command started by downloading the dataset and extracting the *XYZ* information for each molecule. It then arranged the molecules so that their geometric centers were at the beginning of the coordinate system. Then it used the 'xtb tool' (https://github.com/grimme-lab/xtb) to generate a 'molden' file for each molecule, and finally it used ORBKIT (https://orbkit.github.io/) to calculate their corresponding electron densities. This electron densities were calculated for cubes of side 64 units, each unit corresponding to a step size of 0.5 Å. To calculate the electrostatic potentials, the '-esp' flag was sent to 'xtb'. This would return a sparse representation. This sparse representation was placed into an empty cube of sides with 64 units, and the sparse points were dilated to fill a bigger volume. See Supplementary Section 1.3.

## Converting electron densities to SMILES using a transformer model

Our implementation of the transformer architecture followed the standard one as reported by ref. 9. Our encoder, decoder and token embedding followed the standard implementations. The main difference was the embedding layer which inputted the data to the encoder. We called this embedding layer 'molecule embedding'. The aim of this embedding layer was to take as input a 3D tensor representing the electron density of a molecule and outputting a 2D matrix that would operate in the decoder with its 2D attention matrix.

To achieve this transformation from 3D to 2D, first the 3D data were expanded to 4D so that 3D convolutions could be applied. To transform the 4D tensors into 2D, we tested two different strategies.

The first strategy started with 3D convolutions, setting the number of filters to 1, dropping the dimension with size 1 after the convolution had been done, and then repeating this process with 2D convolutions and 1D convolutions until the data were 2D. As an example, if the initial 4D was (64, 64, 64, 64), setting the number of filters to 1 would output (1, 64, 64, 64) and then dropping the first dimension would output (64, 64, 64). If this process is repeated, we would first obtain (1, 64, 64), and then dropping the first dimension we would obtain 2D data (64, 64).

The second strategy used again 3D convolutions, but their strides were of different sizes depending on the dimension. These convolutions were applied until two of the dimensions had a size of 1, and then dropping them, thus getting again 2D data. As an example, if the initial 4D data were (64, 64, 64, 64) and the strides of the 3D convolutions were (1, 2, 2), keeping the number of filters to 64, an initial convolution would output (64, 64, 32, 32). We can repeat these convolutions with these strides until it outputs (64, 64, 1, 1), and then dropping the two single unit dimensions, to obtain (64, 64).

Both strategies produced similar results.

To train the transformer, pairs of (electron density, SMILES) were provided. Note that the electron density could also be the electrostatic potential or decorated electron densities. The electron densities were inputted to the encoder, while the decoder aimed to output the correct SMILES sequence. Once trained, a newer electron density could be inputted to the encoder, while the decoder would receive a start token and output (generate) the corresponding SMILES sequence. See Supplementary Sections 1.4 to 1.12.

## Fitness functions used during the optimization process

The different optimization experiments used a combination of the following fitness functions with different objectives.

- To maximize the size of the molecule.
- To minimize the overlapping between host and guest electron densities.
- To maximize the interactions between host and guest electrostatic potentials.

To perform one step toward maximizing the size of the molecule, the following steps were performed.

(1) Given an input latent vector, the VAE decoder was used to reconstruct the 3D shapes of the molecules.
(2) Tensorflow's 'tf.reduce_sum' took as input the 3D shape and calculated a single value representing the whole 3D electron density by adding together the electron density at each location (within the 64, 64, 64 tensor). This value was used to define the fitness of each molecule.
(3) Tensorflow's 'tf.gradients' calculated the changes needed to increase the fitness of the molecule. This function took as input two parameters: (1) the fitness as just described in the previous point, and (2) the input latent vector. This function (tf.gradients) returned a tensor, which explained how to modify the latent vectors to maximize their fitness values.

To perform one step towards minimizing the overlapping between host and guest electron densities, the sequence of operations was similar to the previous list of operations. The main difference is that in the second step, tf.reduce_sum took as input the product between host and guest. As in this case we wanted to minimize the overlapping, the tensor returned from tf.gradient (in step 3) was subtracted from the latent vectors.

To perform one step toward maximizing the overlapping between host and guest electrostatic potentials, the list of operations was similar to the previous one. The main difference is that now, in the first step, once the VAE generated the electron densities, these electron densities went through the model that generated electrostatic potentials from electron densities (Supplementary Section 1.6). As in this case we wanted to minimize the overlapping, the tensor returned from tf.gradient was subtracted from the latent vectors. For full information, see Supplementary Section 2.1.

To perform a full optimization process, a combination of the previous three fitness functions was used through gradient descent. During each iteration, the latent vectors were modified with the gradient tensor outputted in the third step as discussed before. For full information, see Supplementary Section 2.2.

### Benchmarking the generated SMILES libraries

To benchmark the quality of the molecules generated, nine different sets of molecules were compared (Supplementary Section 4). These 9 sets of 40,000 random latent vectors were generated using a uniform distribution with bounds going from 0.5 up to 50. These latent vectors were then inputted into the VAE decoder to reconstruct their 3D electron densities and electrostatic potentials that were, subsequently, inputted into the transformer model to obtain their SMILES representations. Due to the degeneracy of the SMILES representations generated by our methodology, it was inevitable that duplicate molecules would be obtained. While most of the generated molecules appeared only once or twice, a small fraction of molecules appeared as much as several thousand times, potentially reducing the size of the sets by a quarter after removal of the duplicates. The overall quality of those sets was very high, and almost all SMILES were valid and chemically reasonable (that is, passing structural filters used by popular generators such as MolGen). Around 80% of the molecules were new compared with the training set. Similarity measurements, assessing the similarity between molecules on a scale from zero (different) to one (identical) inside the set of molecules generated (internal) or against the molecules in the training set (external), indicated that the molecules generated were internally diverse and divergent form from the training molecules.

### Cucurbituril CB[6] guest binding titrations

The association constant $K_a$ between **CB[6]** and various amines was determined through $^1$H NMR titration in deuterium oxide ($D_2O$)/formic acid-$d_2$ 1:1, v/v. For each titration, a solution of **CB[6]** with a guest amine was titrated into a solution of the amine, thus maintaining the concentration of the amine constant throughout the titration.

In all **CB[6]**–amine systems, a single set of signals was observed in the $^1$H NMR spectra of the host–guest system, indicating that the system is in fast exchange on the NMR timescale. For each **CB[6]**–amine system, the peak position of a characteristic $^1$H NMR signal of the amine was plotted against the concentration of **CB[6]**. A global nonlinear curve fitting function was then used to fit the data in Origin 2020 to a 1:1 binding model developed by ref. 46.

### Cage [Pd$_2$1$_4$](BArF)$_4$ guest binding titrations

The association constant $K_a$ between [Pd$_2$1$_4$](BArF)$_4$ and various guest molecules was determined through $^1$H NMR titration in dichloromethane-$d_2$ ($CD_2Cl_2$). For each titration, a solution of [Pd$_2$1$_4$](BArF)$_4$ with the studied guest was titrated into a solution of [Pd$_2$1$_4$](BArF)$_4$, thus maintaining the concentration of the cage constant throughout the titration.

In all cage–guest systems, a single set of signals was observed in the $^1$H NMR spectra of the host–guest system, indicating that the system is in fast exchange on the NMR timescale. For each cage–guest system, the peak position of a characteristic $^1$H NMR signal of the pyridine rings of the cage was plotted against the concentration of the guest. A global nonlinear curve fitting function was then used to fit the data in Origin 2020 to the 1:1 binding model developed by ref. 46.

## Data availability

The dataset used to train the models described in this research is the QM9 dataset. This is a publicly available dataset, downloadable from ref. 39. All the data generated through this research are available in the Supplementary Information files. We have also made all the data associated with this work available on Zenodo at https://doi.org/10.5281/zenodo.10530598 (ref. 47). The NMR data used to produce Fig. 6 are available on Zenodo (https://doi.org/10.5281/zenodo.10530598)$^{47}$ and instructions to obtain the binding data is given above in the binding titration sections.

## Code availability

Source code is publicly available at https://github.com/croningp/electrondensity2 and on Zenodo (https://doi.org/10.5281/zenodo.10530598)$^{47}$.

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

## Acknowledgements

We thank S. Pagel for comments on the paper. L.C. gratefully acknowledges financial support from the EPSRC (grant nos. EP/L023652/1, EP/R020914/1, EP/S030603/1, EP/R01308X/1, EP/S017046/1 and EP/S019472/1), the ERC (project no. 670467 SMART-POM), the EC (project no. 766975 MADONNA) and DARPA (project nos. W911NF-18- 2-0036, W911NF-17-1-0316 and HR001119S0003). J.M.G. acknowledges financial support from the Polish National Agency for Academic Exchange grant number PPN/PPO/2020/1/00034 and the National Science Center Poland grant number 2021/01/1/ST4/00007.

## Author contributions

L.C. conceived the concept exploring the direct generation of electron densities. J.M.P.-G., L.W. and J.-F.A. introduced the concept of combining electron density and electrostatic potentials in the 3D representation of molecules. J.M.G. developed the initial models for electron density generation and electron density to SMILES translation. L.W. carried out the preliminary proof of concept of the electron density generation. J.-F.A. selected the guests for experimental testing and realized the experimental work. M.D.B. performed the quantitative analysis of the generated datasets. J.M.P.-G. designed the final artificial intelligence models described in the paper, ran the optimization algorithms and generated the lists of guests. The paper was written by L.C. together with J.M.P.-G. and J.-F.A. with input from all authors.

## Competing interests

The authors declare no competing interests.

## Additional information

**Correspondence and requests for materials** should be addressed to Leroy Cronin.

