## [Peer Review File · Nature Computational Science]

Peer Review Information

Journal: Nature Computational Science

Manuscript Title: Electron Density-Based GPT for Optimization and Suggestion of Host-Guest Binders

Corresponding author name(s): Professor Leroy Cronin

Editorial Notes:

Reviewer Comments & Decisions:

Decision Letter, initial version:
--

Date: 24th August 23 09:57:32

Last Sent: 24th August 23 09:57:32

Triggered By: Kaitlin McCardle

From: kaitlin.mccardle@us.nature.com

To: lee.cronin@glasgow.ac.uk

BCC: kaitlin.mccardle@us.nature.com

Subject: Decision on Nature Computational Science manuscript NATCOMPUTSCI-23-0741-T

Message: ** Please ensure you delete the link to your author homepage in this e-mail if you wish to forward it to your co-authors. **

Dear Professor Cronin,

Your manuscript "Electron Density-Based GPT for Optimization and Suggestion of Host-Guest Binders" has now been seen by 3 referees, whose comments are appended below. You will see that while they find your work of interest, they have raised points that need to be addressed before we can make a decision on publication.

The referees' reports seem to be quite clear. Naturally, we will need you to address all of the points raised.

While we ask you to address all of the points raised, the following points need to be substantially worked on:

- Please add additional quantitative experiments/comparisons, clarifications,

justifications and discussion, as noted by reviewers.

- Please be sure that the language and discussions used throughout the manuscript are accessible to a broad audience

You will also need to make some editorial changes so that it complies with our Guide to Authors at <https://www.nature.com/natcomputsci/for-authors>.

In particular, I would like to highlight the following points of our style:

Nature Computational Science titles should give a sense of the main new findings of a manuscript, and should not contain punctuation. Please keep in mind that we strongly discourage active verbs in titles, and that they should ideally fit within 150 characters each (including spaces).

To improve the accessibility of your paper to readers from other research areas, please pay particular attention to the wording of the paper's abstract, which serves both as an introduction and as a brief, non-technical summary in about 150 words. It should include the background and context of the work, 'Here we show' or an equivalent phrase, and then the major results and conclusions of the paper. Because researchers from other sub-disciplines will be interested in your results and their implications, it is important to explain essential but specialised terms concisely. We suggest you show your summary paragraph to colleagues in other fields to uncover any problematic concepts.

We encourage you to archive the data reported in your manuscript in an accessible, persistent repository. If your data are archived prior to the acceptance of your manuscript, please provide us with the full citation as soon as you receive it so that a link to the data can be included in the publication. See <http://www.nature.com/authors/policies/availability.html> for more information.

If your paper is accepted for publication, we will edit your display items electronically so they conform to our house style and will reproduce clearly in print. If necessary, we will re-size figures to fit single or double column width. If your figures contain several parts, the parts should form a neat rectangle when assembled. Choosing the right electronic format at this stage will speed up the processing of your paper and give the best possible results in print. If you are in doubt about the correct format for your figures after reading our guidelines, please ask the art editors for advice computationalscience@nature.com.

Figure legends must provide a brief description of the figure and the symbols used, including definitions of any error bars employed in the figures.

Please use the following link to submit your revised manuscript and a point-by-point response to the referees' comments (which should be in a separate document to any cover letter):

[REDACTED]

** This url links to your confidential homepage and associated information about manuscripts you may have submitted or be reviewing for us. If you wish to forward this e-mail to co-authors, please delete this link to your homepage first. **

To aid in the review process, we would appreciate it if you could also provide a copy of your manuscript files that indicates your revisions by making use of Track Changes or similar mark-up tools. Please also ensure that all correspondence is marked with your Nature Computational Science reference number in the subject line.

In addition, please make sure to upload a Word Document or LaTeX version of your text, to assist us in the editorial stage.

To improve transparency in authorship, we request that all authors identified as 'corresponding author' on published papers create and link their Open Researcher and Contributor Identifier (ORCID) with their account on the Manuscript Tracking System (MTS), prior to acceptance. ORCID helps the scientific community achieve unambiguous attribution of all scholarly contributions. You can create and link your ORCID from the home page of the MTS by clicking on 'Modify my Springer Nature account'. For more information please visit www.springernature.com/orcid.

We hope to receive your revised paper within three weeks. If you cannot send it within this time, please let us know.

Best regards,

Kaitlin McCardle, PhD
Associate Editor
Nature Computational Science

Reviewers comments:

Reviewer #1 (Remarks to the Author):

In this article, an electron density based GPT model has been developed and host guest optimizations with gradient descent techniques have been performed to discover novel CB[6] and Pd2L4 host guest pairs. The computational favourable guest binding predictions were validated with NMR titration experiments. Overall, I think the article is a difficult read, and the authors should work to make it significantly easier to follow for non-experts.

-In the abstract, it is stated "...host guest binders which are transformed into the SMILES format with >98% accuracy" and in page 4 "... Interestingly, this approach only generated chemically plausible molecules..." and in page 9 "...Due to the degeneracy of the SMILES representations generated by our methodology, it was inevitable that duplicate molecules would be obtained...". The audience is consistently referred to the Supporting Information, yet it is cumbersome to match the critical arguments with the corresponding supporting data since the exact SI numberings are omitted. It should be made clear where each former mentions of SMILES string data comparison corresponds to the state of the data in the analysis steps.

-The electron density image representation of tensor size pruned ((80,80,80) to (64,64,64)) Pd2L4 cage should also be shared.

-How does the total model performance change when different chemical environment descriptors such as SELFIES or DeepSMILES are used for generating the molecules? Would the Variational Autoencoder as Illustrated in Figure 2B be as powerful with a different molecular representation as it is claimed with SMILES strings?

-What is the chemical diversity of the QM9 database and how does direct insertion of this database molecules into the target hosts compare with the candidate molecules? Larger heavy atom count libraries such as QM7 database seems to be more suitable for the Pd2L4 host as there is available free volume within the host.

-It is unclear from the main text and the SI which xTB version and which parameters are selected for electrostatic calculations. xTB methods may sometimes give erroneous calculation results if the initial geometry of the system is not valid. How does the xTB method electron density files for the most promising guest molecules (that were generated) compare with electron densities from pairing DFT calculations?

-How transferrable is just placing the guest molecule at the centre of the host? Would inclusion of angular rotation parameters enhance and direct the results and let the ESP maps of host and guest match better?

-Why isn't slight overlap of electron densities considered? The candidate guest molecules can have differing degrees of flexibility as illustrated in the number of double, triple bonds section of the SI, as well as the host molecule due to thermal vibrations. A single molecule per host is assumed so this relaxation of the initial placement can potentially add an ensemble of candidates with one more heavier atom.

-In Figure 8, the association constants are weaker than for those guests that are previously known (on the left hand side). Could the authors do a general comparison of this point?

-In the abstract, the "discover" should be "discovery" in the last sentence.

I am reassured to see the authors will publish their model on github, it is important that this is indeed done for any accepted version.

Reviewer #2 (Remarks to the Author):

1. Given there are a lot of approaches that are focused on target-aware drug design based on electron densities [1-4], I think it is necessary to add some comparisons with them, for example, Targetdiff[1] and DecompDiff[4].

2. The paper's current approach only presents a limited view of its results, as it solely showcases a few desirable guests selected by an expert chemist. This approach lacks

objectivity and is insufficient to assess the overall performance of the generated molecules before expert intervention. To address this, the paper should consider adopting an evaluation step similar to the one used in the DecompDiff paper, which provides a more comprehensive assessment of the generated results. Furthermore, I suggest conducting an ablation study on the major components of the pipeline. For instance, the paper could demonstrate the performance of generated guests without utilizing the optimization pipeline. This addition would be valuable for readers, as it would clearly highlight the individual contributions of each novel idea in the molecule generation process.

[1]: Guan, Jiaqi, et al. "3d equivariant diffusion for target-aware molecule generation and affinity prediction." arXiv preprint arXiv:2303.03543 (2023).

[2]: Peng, Xingang, et al. "Pocket2mol: Efficient molecular sampling based on 3d protein pockets." International Conference on Machine Learning. PMLR, 2022.

[3]: Luo, Shitong, et al. "A 3D generative model for structure-based drug design." Advances in Neural Information Processing Systems 34 (2021): 6229-6239.

[4]: Guan, Jiaqi, et al. "DecompDiff: Diffusion Models with Decomposed Priors for Structure-Based Drug Design." (2023).

Reviewer #2 (Remarks on code availability):

The authors have generously provided the source code, a list of dependent packages, and the trained model for their paper. I haven't had the opportunity to install and run it yet.

Reviewer #3 (Remarks to the Author):

Review on the article titled:

"Electron Density-Based GPT for Optimization and Suggestion of Host-Guest Binders"

This manuscript presents a new approach towards generative modeling of molecules using host-guest systems as an example. In this method, representing the host molecules with their electron density decorated with electrostatic potential, new guest molecules are generated with improved host-guest interaction, by maximizing the inter-molecular interactions and minimizing the overlap with the host. The overall workflow is two-tiered: first, virtual libraries of potential guest molecules are generated; in this step, first, electron density volumetric representation of new molecules are generated and optimized by minimizing the overlap and maximizing the electrostatic interaction and second, potential guest molecules are selected for in-vitro testing of binding affinity.

The language in the manuscript is lucid and easy to read. The authors discussed their findings thoroughly, however, here are a few suggestion/comments that may help in improving the readability of the manuscript:

In Page 8: While explaining "Translating electron densities into SMILES", the author mentioned 3D to 4D expansion of the tensor data in this line "To do so, the input 3D data first had to be transformed and expanded into 4D— enabling 3D convolutions— before this 4D data was transformed into 2D. "

While the steps are clearly explained, the reason behind this 4D expansion needs a little more explanation. It would be very useful if the authors explain in detail the reason behind expanding the electron density tensors to add another dimension.

In the section "Quantitative study of the host-guest recognition": the authors studied in-vitro binding affinity tests on two different host-guest systems. The authors described in detail about the affinities of the new guest molecules, but if the authors could provide the rationale behind selecting 9 guests for Cucurbituril and 4 from the metal-organic cage host from the generative workflow, which is not mentioned in the current version of the manuscript.

In Page 14, 15: the authors presented the selected guests for two host-guest systems in Fig. 7 and 8 respectively. For a better visualization and presentation of the binding affinities, one of the suggestions would be to convert these figures into tables for known and new guests.

Minor corrections for typos, if applicable:

In the Abstract, the last line, "discovery" instead of "discover"

Page 7: "fitting the closest these volumes" to "fitting the closest to these volumes"

Page 11 : valued to "values"

Author Rebuttal to Initial comments

We would like to thank the editors for considering our paper for publication in Nature Computational Science as well as the reviewers for their constructive feedback. We apologize for the delay in responding to the reviewers' comments - they have inspired us to perform some new experiments that took time to develop.

Please find below our point-by-point answer to the reviewers (replies in italics).

Reviewer #1:

In this article, an electron density based GPT model has been developed and host guest optimizations with gradient descent techniques have been performed to discover novel CB[6] and Pd2L4 host guest pairs. The computational favourable guest binding predictions were validated with NMR titration experiments.

Firstly, we would like to thank Reviewer 1 for taking their time to read our paper and for their insightful comments. We have addressed all the comments, including the development of the new experiments, and we think that with the new data obtained according to the suggestions of the reviewer, we have strengthened the manuscript.

Overall, I think the article is a difficult read, and the authors should work to make it significantly easier to follow for non-experts.

This reviewer suggested to make changes to the manuscript to make it easier to follow for non-experts. However, reviewer 3 said that the paper is "lucid and easy to read". Given the mixed feedback we received from the reviewers in this regard, and the early stage of editorial process, we opted to introduce only modest modifications to the manuscript. We are keen to follow any suggestions on this matter.

We believe that the paper is easy to read for non-experts. Notably, the main two authors who wrote the article come from very diverse backgrounds: one of them is purely an organic synthetic chemist with minimal knowledge about programming and AI, while the other is a computer scientist without any knowledge of chemistry. We wrote the paper collaboratively to make sure that we were able to understand the parts written by the other coauthor. The paper was also iterated with other members of our research group, from very different disciplines, who provided feedback about how to improve it and make it readable.

-In the abstract, it is stated "...host guest binders which are transformed into the SMILES format with >98% accuracy" and in page 4 "... Interestingly, this approach only generated chemically plausible molecules..." and in page 9 "...Due to the degeneracy of the SMILES representations generated by our methodology, it was inevitable that duplicate molecules would be obtained...". The audience is consistently referred to the Supporting Information, yet it is cumbersome to match the critical arguments with the corresponding supporting data since the exact SI numberings are omitted. It should be made clear where each former mentions of SMILES string data comparison corresponds to the state of the data in the analysis steps.

References to specific sections of the supporting information have been added to the main text to guide the reader.

-The electron density image representation of tensor size pruned ((80,80,80) to (64,64,64)) Pd2L4 cage should also be shared.

To prune the tensor we simply sliced the Python array:

Host = host[:,8:-8,8:-8,8:-8,:]

This information as well as the images of the host before and after pruning have been added to the Supplementary Information (Section 2.1.4).

Python's Pickle binary files of the two hosts have been added to the Git Repository.

-How does the total model performance change when different chemical environment descriptors such as SELFIES or DeepSMILES are used for generating the molecules?

Following the suggestion from the reviewer, we have retrained the full workflow of molecule generation using SELFIES instead of SMILES. This involved re-generating the full dataset and re-training all the building blocks. We have added the new results with SELFIES to the Supplementary Information, Sections 2.3.10 and 2.3.11 (around 70 new pages of new content and almost 3,000 new guests).

The big advantage of using SELFIES instead of SMILES is that all the sequences of tokens it generates are valid – meaning that every generated sequence represents a valid molecule. This can be seen in the obtained results, as there are no gaps in the presented tables, as opposed to what happened with SMILES. Nevertheless, when the results were screened by an expert chemist, they did not contain any new potential molecule (as in the molecule being a known guest or similar a known one) that were not already present in the batches of molecules generated using SMILES. Based on these results, we considered that both SMILES and SELFIES offered similar results.

We have added this information to the main manuscript.

Would the Variational Autoencoder as Illustrated in Figure 2B be as powerful with a different molecular representation as it is claimed with SMILES strings?

The Variational Autoencoder does not work with text representation of molecules (SMILES or SELFIES). It takes the 3D volumes representing Electron Densities or Electrostatic Potentials as input/output. Therefore, at this specific stage of our pipeline the molecular representation is irrelevant.

-What is the chemical diversity of the QM9 database and how does direct insertion of this database molecules into the target hosts compare with the candidate molecules?

The chemical diversity of the QM9 database has been widely studied before, for example in

Glavatskikh, M., Leguy, J., Hunault, G. *et al.* Dataset's chemical diversity limits the generalizability of machine learning predictions. *J Cheminform* **11**, 69 (2019). <https://doi.org/10.1186/s13321-019-0391-2> (The section named "Chemical differences between the QM9 and PC9" contains very complete information about the chemical diversity of the QM9 dataset).

Nevertheless, even though some newer and "more diverse" datasets exists, such as the PC9 discussed in the linked paper, the QM9 was chosen for this research as it is considered the golden standard for Machine Learning predictions of various chemical properties, and it perfectly fitted our target task (host-guest optimisations).

Our Supplementary Information, in the Benchmark section, contains a complete analysis of the chemical diversity of the molecules generated using our AI models. At the start of the Section 4, for example, it is described that depending on the initial random vector generation processes as well as on the sampling procedure, up to 70% of the generated molecules were unique. Similarly, during benchmarking up to 80% of the molecules generated by our AI were not present in the QM9 dataset. These results show that our AI models are generating new molecules and not just reusing molecules already present in the QM9 dataset.

Nevertheless, the reviewer has raised an excellent point. We have curated a list of known guests for the CB6 and the Pd Host (please see Supplementary Information Section 4.5). Out of the 52 guests for CB6 described in the literature, only 3 are found in the QM9 dataset. While for the Pd Cage, only 4 out of the 14 hosts described in the literature are present in the QM9 dataset. These results show that the QM9 dataset does not contain, by default, a high quantity of good guests for our host molecules.

If we focus on the guests discovered by our AI, only 1 out of the 16 guests presented for the CB6 is found in the QM9 dataset. While out of the 8 guests present for the Pd Cage, 4 of them were present in the QM9

dataset. Therefore, 19 out of the 24 guests presented in the main manuscript were new and are not found in the QM9 dataset.

All this information has been added to Supplementary Information Section 4.5.

Larger heavy atom count libraries such as QM7 database seems to be more suitable for the Pd2L4 host as there is available free volume within the host.

To the best of our knowledge, the QM7 dataset contains smaller molecules (up to 7 heavy atoms). While the QM9 dataset (the one we used) contains bigger molecules with up to 9 heavy atoms.

Nevertheless, we fully agree that the molecules from the QM9 are small for the Pd Cage. This is why, when optimising for this host, our optimisation algorithm tried to make the molecules as big as possible (as well as minimising overlapping and maximising electrostatic interactions). It is our objective in future research to use a database of bigger molecules. One of the reasons why the QM9 dataset was chosen was because it contains a total of 130,000 molecules, and we could fit them in 64,64,64 tensors. This enabled us to run the AI algorithms in a decent time-frame using our hardware (a combination of RTX A6000, RTX 2080TI and RTX TITAN). With this hardware, as an example, training the VAE took around 3 weeks. A bigger dataset like the GDB17 with millions of bigger molecules might take months to train.

Using a database with bigger molecules is an excellent suggestion, and we have added some comments about this in the discussion section of the main manuscript.

-It is unclear from the main text and the SI which xTB version and which parameters are selected for electrostatic calculations. xTB methods may sometimes give erroneous calculation results if the initial geometry of the system is not valid. How does the xTB method electron density files for the most promising guest molecules (that were generated) compare with electron densities from pairing DFT calculations?

The version of xTB used is 6.4.1 (this was the latest available version when this research started). The parameters used were:

```
"xtb --model --esp"
```

xTB was only used when creating the dataset to calculate the electron densities and electrostatic potentials. xTB calculations were never used during the generative process and host-guest optimisations.

To generate the electron densities we also used Orbkkit (git checkout version cc170172).

Full information about how the electron densities and electrostatic calculations were generated can be seen in Section 1.3 from the Supplementary Information. The supplementary information also points to the different files and functions used within the source code.

-How transferrable is just placing the guest molecule at the centre of the host? Would inclusion of angular rotation parameters enhance and direct the results and let the ESP maps of host and guest match better?

The guests were placed at the centre of the host. More precisely, both guests and hosts were centered at position 0,0,0.

During the training phase we used data augmentation, namely, the guests were randomly rotated in each direction before being inputted to the model. This rotation was done individually for each molecule at every epoch. Meaning that during the whole training phase, the same molecule was presented in different rotations at every epoch to make the model learn different poses of molecules.

The optimisation algorithms modified the guests in the latent space to increase the fitness function. For certain cases this optimisation might lead to different operations, for example adjusting the size and shape as well as rotation of the generated guest molecules. In our opinion, the model presented should have the

capacity to be rotation invariant to certain extent, thus, we think that the inclusion of an angular rotation parameter would have minimal effect on the generative process.

-Why isn't slight overlap of electron densities considered? The candidate guest molecules can have differing degrees of flexibility as illustrated in the number of double, triple bonds section of the SI, as well as the host molecule due to thermal vibrations. A single molecule per host is assumed so this relaxation of the initial placement can potentially add an ensemble of candidates with one more heavier atom.

Slight overlaps of electron densities indeed were considered and even "forced" by our optimisation pipeline.

The first step of our optimisation pipeline consisted on making the molecules as big as possible in order to force as much overlapping as possible.

The optimisation pipeline was followed by two steps that were executed together many times in a sequence. The first step aimed to minimise the overlap of the electron densities between the guest and the host, while the second step aimed to maximise the electrostatic interactions between host and guest.

While the first step aimed to minimise the overlapping of electron densities, in practice, an overlapping of 0 (i.e., no overlapping at all) was never achieved. Moreover, since our gradient descent used the overlapping factor to change the latent vectors, an overlap of 0 was never desirable in the first place, because it prevented any further adjustments from the algorithm.

The second step aimed to maximise the electrostatic interactions, which in practice meant generating more diffuse electron densities of guest molecules, thereby more overlapping of electron densities.

In summary, the overlap between molecules is what drove our optimization algorithms, so we always wanted to maintain a nominal overlap between host and guest.

-In Figure 8, the association constants are weaker than for those guests that are previously known (on the left hand side). Could the authors do a general comparison of this point?

The section b of "Quantitative study of the host-guest recognition" has been modified to address the comment of the referee:

"In all four cases (see Figure 8), the affinity of the guest for $[\text{Pd}_2\mathbf{1}_4](\text{BARF})_4$ was in line with the lower range of affinities previously reported for "small-sized neutral guests" in CD_2Cl_2 (i.e., guest formed of 10 heavy atoms or less, such as \mathbf{G}^{19}) (36). The lack of "strong binders" in the molecules tested could be attributed to the fact that the cavity size of $[\text{Pd}_2\mathbf{1}_4](\text{BARF})_4$ pushes the limits of our model and workflow capabilities: (i) as previously highlighted, the scarcity of commercially available options within the dataset generated by our model hampered the quality of the guest tested, (ii) all known strong binders for $[\text{Pd}_2\mathbf{1}_4]^{4+}$ feature an aromatic core substituted by two donor groups in para to each other (35, 36). Apart from \mathbf{G}^{17} , this structural feature inherently increases the size of the molecule beyond the 10 heavy atoms limit of our model. Such size constraint on the molecules generated by our model stems from the utilization of QM9 for its training, making it unlikely to generate molecules that exceed 10 heavy atoms in size."

-In the abstract, the "discover" should be "discovery" in the last sentence.

The text has been modified accordingly.

I am reassured to see the authors will publish their model on github, it is important that this is indeed done for any accepted version.

We can confirm that all the source code will be freely available on GitHub after manuscript publication.

Reviewer #2:

1. Given there are a lot of approaches that are focused on target-aware drug design based on electron densities [1-4], I think it is necessary to add some comparisons with them, for example, Targetdiff[1] and DecompDiff[4].

We are very grateful to the reviewer for recommending these excellent papers from Jian Peng and Juanzhu Ma. We have added references to them in our manuscript. Nevertheless, we think that our approach is very different the one suggested in these papers:

First of all, the subset of chemistry is substantially different. The suggested references focus on drug design and modeling interactions with proteins, while ours work is focused on host-guest chemistry in supramolecular chemistry.

Secondly, the main outcome we achieved in our research is the actual experimental validation of the discovered molecules. To demonstrate this, we performed an additional step in which the binding between the suggested supramolecular guests and the hosts was experimentally tested. This additional step heavily influenced our workflow. A potential drawback of many generative models in the field of cheminformatics is that the molecules generated by the AI models do not perform well in real experiments. See, for example, the following research that analyses the results from AlphaFold: <https://www.biorxiv.org/content/10.1101/2022.11.21.517405v1>

Finally, the way our research used electron densities is very different to the research done by Jian Peng and Juanzhu Ma. Their research (such as <https://arxiv.org/pdf/2303.03543.pdf>) uses graph neural networks (SE(3)-equivariant) to place atoms in the protein pocket, based on probability densities (for example using a diffusion model). Our approach considers the guest and the cavity of the host as fully 3D volumes, and it manipulates the 3D volume of the guest to maximise the fitness functions. Only when the optimization algorithm finished, we passed the 3D volume through a Transformer model, and we assigned different atoms and bonds to different parts of the volume to finally obtain a molecule. A key element of our research was to work with electron densities represented purely as 3D volumes – without any kind of atom information-, and to manipulate these 3D volumes to maximise the different fitness functions. We consider that this is very different to the papers suggested by this reviewer.

Because our approach is very different, it is not possible to compare it directly with Targetdiff or DecompDiff models which were using as the training dataset, for example, Docked2020 dataset, while we used the QM9 dataset. The main benchmark of their comparisons is based on the Vina Score, which is a scoring function for protein binding, but that has no meaning in the subset of chemistry we focused on. Some of the values, such as QED, SA and Diversity could be compared. Nevertheless due to significant disparities between these datasets any such comparisons may lack substantial significance. To ensure transparency, we have provided full information about this, as well as other benchmarking qualities, in the Benchmark section within our Supplementary Information.

In our research the primary objective was to compare our results with known guests synthesised by expert chemists. Our objective was to create an AI system that excels in real-world practical chemistry.

2. The paper's current approach only presents a limited view of its results, as it solely showcases a few desirable guests selected by an expert chemist. This approach lacks objectivity and is insufficient to assess the overall performance of the generated molecules before expert intervention.

The main manuscript contains a limited view of the results due to space constraints. The accompanying Supplementary Information contains almost 200 pages of AI-generated guests, generated under a wide variety of conditions. In total, our Supplementary Information displays over 75,000 guests. For clarity purposes as well as to make the main manuscript more friendly to the reader, we decided to leave these results in the Supplementary Information. If it is deemed beneficial or necessary, we are more than willing to include some of these results to the main manuscript.

To address this, the paper should consider adopting an evaluation step similar to the one used in the DecompDiff paper, which provides a more comprehensive assessment of the generated results.

We fully agree with this reviewer that adopting an evaluation step that by-passes the current human-screening is one of the best ways to improve the presented research. Nevertheless, we consider that this idea is out of the scope for the presented research, and it would entail an entirely different project. We have added comments about this in the discussion section of the main manuscript.

Furthermore, I suggest conducting an ablation study on the major components of the pipeline. For instance, the paper could demonstrate the performance of generated guests without utilizing the optimization pipeline. This addition would be valuable for readers, as it would clearly highlight the individual contributions of each novel idea in the molecule generation process.

The Supplementary Information already contains this. Section 4: Benchmarking, contains around 70 pages and it conducts a Benchmark study of the guests generated without utilising the optimization pipeline.

[1]: Guan, Jiaqi, et al. "3d equivariant diffusion for target-aware molecule generation and affinity prediction." *arXiv preprint arXiv:2303.03543* (2023).

[2]: Peng, Xingang, et al. "Pocket2mol: Efficient molecular sampling based on 3d protein pockets." *International Conference on Machine Learning*. PMLR, 2022.

[3]: Luo, Shitong, et al. "A 3D generative model for structure-based drug design." *Advances in Neural Information Processing Systems* 34 (2021): 6229-6239.

[4]: Guan, Jiaqi, et al. "DecompDiff: Diffusion Models with Decomposed Priors for Structure-Based Drug Design." (2023).

Reviewer #2 (Remarks on code availability):

The authors have generously provided the source code, a list of dependent packages, and the trained model for their paper. I haven't had the opportunity to install and run it yet.

Based on the comments from the referees, a new version of the source code has been made available accompanying the main manuscript.

Reviewer #3:

Review on the article titled: "Electron Density-Based GPT for Optimization and Suggestion of Host-Guest Binders"

This manuscript presents a new approach towards generative modeling of molecules using host-guest systems as an example. In this method, representing the host molecules with their electron density decorated with electrostatic potential, new guest molecules are generated with improved host-guest interaction, by maximizing the inter-molecular interactions and minimizing the overlap with the host. The overall workflow is two-tiered: first, virtual libraries of potential guest molecules are generated; in this step, first, electron density volumetric representation of new molecules are generated and optimized by minimizing the overlap and maximizing the electrostatic interaction and second, potential guest molecules are selected for in-vitro testing of binding affinity.

The language in the manuscript is lucid and easy to read. The authors discussed their findings thoroughly, however, here are a few suggestion/comments that may help in improving the readability of the manuscript:

We would like to thank the referee for taking their time to review our paper.

In Page 8: While explaining “Translating electron densities into SMILES”, the author mentioned 3D to 4D expansion of the tensor data in this line “To do so, the input 3D data first had to be transformed and expanded into 4D— enabling 3D convolutions—before this 4D data was transformed into 2D. “

While the steps are clearly explained, the reason behind this 4D expansion needs a little more explanation. It would be very useful if the authors explain in detail the reason behind expanding the electron density tensors to add another dimension.

In order to perform 3D convolutions, the input data must be 4D. This is a requirement set by the AI library we used (Tensorflow).

We have added information about this in the main manuscript.

In the section “Quantitative study of the host-guest recognition”: the authors studied in-vitro binding affinity tests on two different host-guest systems. The authors described in detail about the affinities of the new guest molecules, but if the authors could provide the rationale behind selecting 9 guests for Cucurbituril and 4 from the metal-organic cage host from the generative workflow, which is not mentioned in the current version of the manuscript.

The section b of “Quantitative study of the host-guest recognition” has been modified to address the comment of the referee:

“Compared to **CB[6]**, featuring a cavity with a diameter of approximately 3.9 Å (30), **[Pd₂1₄]⁴⁺** possesses a notably larger cavity, measuring approximately 6 Å in width and 10 Å in depth (35). This increased cavity size led our model to generate larger guest molecules, resulting in very few of them being commercially available thereby limiting the pool of molecules available for experimental testing. “

In Page 14, 15: the authors presented the selected guests for two host-guest systems in Fig. 7 and 8 respectively. For a better visualization and presentation of the binding affinities, one of the suggestions would be to convert these figures into tables for known and new guests.

Figures 7 and 8 have been modified accordingly.

Minor corrections for typos, if applicable:

In the Abstract, the last line, “discovery” instead of “discover”

Page 7: “fitting the closest these volumes” to “fitting the closest to these volumes”

Page 11 : valued to “values”

The text has been modified accordingly.

Decision Letter, first revision:

Date: 7th November 23 07:09:03
Last Sent: 7th November 23 07:09:03
Triggered By: Kaitlin McCardle
From: kaitlin.mccardle@us.nature.com
To: lee.cronin@glasgow.ac.uk
CC: computacionalscience@nature.com
BCC: kaitlin.mccardle@us.nature.com
Subject: AIP Decision on Manuscript NATCOMPUTSCI-23-0741A
Message: Our ref: NATCOMPUTSCI-23-0741A

7th November 2023

Dear Dr. Cronin,

Thank you for submitting your revised manuscript "Electron Density-Based GPT for Optimization and Suggestion of Host-Guest Binders" (NATCOMPUTSCI-23-0741A). It has now been seen by the original referees and their comments are below. The reviewers find that the paper has improved in revision, and therefore we'll be happy in principle to publish it in Nature Computational Science, pending minor revisions to satisfy the referees' final requests and to comply with our editorial and formatting guidelines.

TRANSPARENT PEER REVIEW

Nature Computational Science offers a transparent peer review option for original research manuscripts. We encourage increased transparency in peer review by publishing the reviewer comments, author rebuttal letters and editorial decision letters if the authors agree. Such peer review material is made available as a supplementary peer review file. **Please remember to choose, using the manuscript system, whether or not you want to participate in transparent peer review.**

Thank you again for your interest in Nature Computational Science. Please do not hesitate to contact me if you have any questions.

Sincerely,

Kaitlin McCardle, PhD
Senior Editor
Nature Computational Science

ORCID

Reviewer #1 (Remarks to the Author):

The manuscript can now be accepted in my opinion. There are a few small typos to be corrected at proof stage.

Reviewer #1 (Remarks on code availability):

It is difficult to find the rotation element within the code repository. The rotation element should live inside `bin/optimisers/host_guest_overlapping.py` as shown in SI 2.2.1. Yet one has to dive into `src/utils/optimiser_utils.py` to find elements of rotation within the code, and it is not well documented how to act upon this element. (No user comments) Also, the data paths are not properly formatted to target correct user independent subdirectories (such as `"DATA_FOLDER = '/home/juanma/Data/' # in maddog2020"` in line 42 `host_guest_overlapping.py` and comments should be refined to help users with functionalities. Distribution of the code also through either a docker container or a google collab binder document is potentially better as these approaches will not force the community to download 250 GB+ raw QM9 data.

Reviewer #2 (Remarks to the Author):

Thanks for the explanation and revision!

1. Sorry for the confusion. I didn't believe your method was quite similar to those approaches. I just thought these two tasks were similar. It is because when considering computational model or framework design, it appears that designing drugs based on pockets and designing guest molecules based on the host are quite similar. Therefore, I initially believed that it would be beneficial to draw a comparison between methods under these two tasks. However, I acknowledge your assertion that they are indeed distinct tasks within the realm of chemistry, so it is not necessary to compare them.

2.3.4. I appreciate your reminder about the supplementary information and apologize

for not considering it in the initial review. Upon reviewing the results provided in the supplementary information, I find the evaluation to be more comprehensive.

Nonetheless, I still believe that displaying the overall performance prior to expert intervention would be very valuable, as it represents an unbiased way to directly showcase your method's effectiveness. However, I understand that if your algorithm is the first in silico design method in the field of guest molecule design, it may be acceptable to omit these preliminary results.

I recommend incorporating a concise summary of the results from the supplementary information into the main text. I believe that would be beneficial for the reader.

Reviewer #2 (Remarks on code availability):

Yes, I successfully installed their package and their pretrained model could be successfully loaded.

Reviewer #3 (Remarks to the Author):

I would like to thank the authors for carefully reading the comments and suggestions and making necessary changes or add the required information to the manuscript.

Final Decision Letter:

Date: 23rd January 24 13:27:15

Last Sent: 23rd January 24 13:27:15

Triggered By: Kaitlin McCardle

From: kaitlin.mccardle@us.nature.com

To: lee.cronin@glasgow.ac.uk

BCC: kaitlin.mccardle@us.nature.com,fernando.chirigati@us.nature.com,computationalscience@nature.com,risproduction@springernature.com

Subject: Decision on Nature Computational Science manuscript NATCOMPUTSCI-23-0741B

Message: Dear Professor Cronin,
:

We are pleased to inform you that your Article "Electron Density-Based GPT for Optimization and Suggestion of Host-Guest Binders" has now been accepted for publication in Nature Computational Science.

Once your manuscript is typeset, you will receive an email with a link to choose the appropriate publishing options for your paper and our Author Services team will be in touch regarding any additional information that may be required.

Please note that *Nature Computational Science* is a Transformative Journal (TJ). Authors may publish their research with us through the traditional subscription access route or make their paper immediately open access through payment of an article-processing charge (APC). Authors will not be required to make a final decision about access to their article until it has been accepted. [Find out more about Transformative Journals](https://www.springernature.com/gp/open-research/transformative-journals)

Authors may need to take specific actions to achieve [compliance with funder and institutional open access mandates](https://www.springernature.com/gp/open-research/funding/policy-compliance-faqs). If your research is supported by a funder that requires immediate open access (e.g. according to [Plan S principles](https://www.springernature.com/gp/open-research/plan-s-compliance)) then you should select the gold OA route, and we will direct you to the compliant route where possible. For authors selecting the subscription publication route, the journal's standard licensing terms will need to be accepted, including [self-archiving policies](https://www.springernature.com/gp/open-research/policies/journal-policies). Those licensing terms will supersede any other terms that the author or any third party may assert apply to any version of the manuscript.

Acceptance of your manuscript is conditional on all authors' agreement with our publication policies (see <https://www.nature.com/natcomputsci/for-authors>). In particular your manuscript must not be published elsewhere and there must be no announcement of the work to any media outlet until the publication date (the day on which it is uploaded onto our web site).

Before your manuscript is typeset, we will edit the text to ensure it is intelligible to our wide readership and conforms to house style. We look particularly carefully at the titles of all papers to ensure that they are relatively brief and understandable.

Once your manuscript is typeset, you will receive a link to your electronic proof via email with a request to make any corrections within 48 hours. If, when you receive your proof, you cannot meet this deadline, please inform us at rjsproduction@springernature.com immediately.

If you have queries at any point during the production process then please contact the production team at rjsproduction@springernature.com.

An online order form for reprints of your paper is available at a

<https://www.nature.com/reprints/author-reprints.html>><https://www.nature.com/reprints/author-reprints.html>. All co-authors, authors' institutions and authors' funding agencies can order reprints using the form appropriate to their geographical region.

We welcome the submission of potential cover material (including a short caption of around 40 words) related to your manuscript; suggestions should be sent to Nature Computational Science as electronic files (the image should be 300 dpi at 210 x 297 mm in either TIFF or JPEG format). We also welcome suggestions for the Hero Image, which appears at the top of our <http://www.nature.com/natcomputsci>>home page; these should be 72 dpi at 1400 x 400 pixels in JPEG format. Please note that such pictures should be selected more for their aesthetic appeal than for their scientific content, and that colour images work better than black and white or grayscale images. Please do not try to design a cover with the Nature Computational Science logo etc., and please do not submit composites of images related to your work. I am sure you will understand that we cannot make any promise as to whether any of your suggestions might be selected for the cover of the journal.

Best regards,

Kaitlin McCardle, PhD
Senior Editor
Nature Computational Science

P.S. Click on the following link if you would like to recommend Nature Computational Science to your librarian: <https://www.springernature.com/gp/librarians/recommend-to-your-library>><https://www.springernature.com/gp/librarians/recommend-to-your-library>

** Visit the Springer Nature Editorial and Publishing website at <http://editorial-jobs.springernature.com>>[www.springernature.com/editorial-and-publishing-jobs](http://editorial-jobs.springernature.com) for more information about our career opportunities. If you have any questions please click [here](mailto:editorial.publishing.jobs@springernature.com).**